# AR4FSM: Mobile Augmented Reality Application in Engineering Education for Finite-State Machine Understanding

**Muhammad Nadeem** [1],*[iD], **Mayank Lal** [2], **Jiaming Cen** [2] and **Mohammad Sharsheer** [1]

1. College of Engineering and Technology, American University of the Middle East, Egaila 54200, Kuwait
2. Department of Electrical, Computer, and Software Engineering, University of Auckland, Auckland 1010, New Zealand
* Correspondence: muhammad.nadeem@aum.edu.kw

**Abstract:** Students in the twenty-first century are accustomed to using technology in all aspects of their lives and have never known a world without it; the classroom is no exception. Augmented reality (AR) is a technology that bridges the virtual and physical worlds to make learning more engaging and enjoyable. In this paper, we present a mobile application aimed at novice learners that makes use of technology for the teaching and learning of computer system engineering concepts. Currently, students typically learn about finite-state machine (FSM) concepts from lectures, tutorials, and practical hands-on experience combined with commercial timing simulation tools. We aimed to enhance these traditional, lecture-based instruction and information delivery methods. We developed an AR-based FSM visualization tool called AR4FSM to help students more easily grasp concepts through immersion and natural interaction with an FSM. We used a blend of multimedia information, such as text, images, sound, and animations superimposed on real-world-state machine diagrams, presenting the information in an interactive and compelling way. An experiment with 60 students showed that the app was perceived positively by the students and helped to deliver FSM-related concepts in a way that was easier to understand than traditional, lecture-based teaching methods. This instruction methodology not only engaged the students but also motivated them to learn the material. The findings of this study have inspired us to use this application to teach FSM topics in the classroom.

**Keywords:** active learning; augmented reality; educational technology; engagement; finite-state machine; instruction methods; mobile applications; teaching and learning

## 1. Introduction

The current generation of university entrants is very well conversant with the use of multimedia technology, which is largely the result of the tremendous growth in communications and information technology in recent years. Technologies have become indispensable components of their lives, which has also significantly changed the way they acquire new knowledge. Although outdated teaching methods and paradigms are still being enforced and seem to be working, most universities are interested in bringing about a change. They are already benefiting from implementing newly available multimedia technologies to improve instruction methods with the goal of creating deep learning and explaining complex concepts in an easier way. This is also recommended by practitioners and researchers who argue that blending technological learning solutions with traditional classroom practice can create a better learning environment [1,2]. Additionally, Mayer's cognitive theory of multimedia learning (CTML) proposes that the brain does not view a multimedia presentation of words, visuals, and aural information in a mutually exclusive manner; rather, these elements are dynamically chosen and structured to build logical mental constructions [3]. This suggests that human beings can use two processing channels during the learning process, and both are maneuvered in active processing, but each individual channel suffers

from capacity limitation. Therefore, it is recommended to leverage two channels in multimedia representation as it allows us to process more information than a single channel. This means the instruction material which includes different representation modes, such as text and graphics, or different sensory modes, such as visuals and auditory, can be used effectively to enhance the learning process, as described by the modality principles [4]. This can be extended to different realities, such as physical and virtual.

Recently, virtual technology has become one of the most popular trends in daily life [5]. Virtual reality (VR) provides a simulated experience which can be different from the physical world. AR goes a step further and amalgamates virtual, digital contents with real-world contents in a synchronized way. This bridging of the gap between the virtual and physical world can replace memory-based learning with a more fun-driven way of learning, thus fostering more conceptual and meaningful learning, as argued by John Biggs [6]. Web 2.0 tools, such as blogs, wikis, and multi-user virtual environment (MUVE), have been integrated into teaching and learning to make classrooms more active through content sharing and idea collaboration. AR and 3D Virtual Worlds (3DVWs) provide an immersive environment by enabling the perception of objects from different perspectives simultaneously and inspiring students to interact, which was not possible with earlier technologies [7]. Though newer technologies offer many advantages, they have downsides as well. The teacher is often reluctant to embrace newer technology due to a lack of skills, and this inculcates poor teaching habits among the students [8]. However, the benefits offered by AR outweigh the disadvantages, as the main feature of AR is that it is driven by the learning-by-doing paradigm.

Since its introduction as a training tool for airline and Air Force pilots during the 1990s [8], AR is now widely used in many fields, including medicine [9], rehabilitation [10], lab orientation [11], tourism [12], publicity [13], training [14], and education [15]. It is used in STEM education more frequently nowadays [16]. Several studies reported the effectiveness of the use of AR for better learning performance and motivation, student engagement, facilitating interaction and collaboration, providing just-in-time information, creating situated learning environments, and increasing the capacity for innovation [17]. The use of AR for teaching fosters active engagement with students, which results in high-quality teaching and learning [18]. It also allows the students to take control of their learning, saving the teacher's time, which was originally spent repeating explanations.

In engineering education, the use of visual representations and physical models is very important, as their absence makes it hard to grasp difficult concepts in an abstract way and means students must rely on mere imagination. AR is particularly helpful in such cases, and it is used in electrical engineering for explaining concepts related to magnetism [19], electricity [20], and antenna waves [21].

FSM is one of the most essential topics in digital electronics because it provides a formal methodology for a designer to translate the specification of a digital control circuit to actual circuits. Therefore, it is extremely important that the student fully understands the working principles of FSMs in order to implement digital systems. However, it is not possible to easily observe the behavior of a FSM using state tables or waveforms generated by simulation tools. State diagram representation is easier to grasp, but its static nature makes it hard for students to observe the workings and verify the functionality. A visual simulation, while keeping intact the simple and easily observable state diagram representation of the FSM, makes it easier to understand the functionality and principles of the FSM. Furthermore, AR is very cheap to use and does not involve many occupational risks. The current study solves this problem through the design and development of an AR application, AR4FSM, that provides students with an opportunity to interact with the graphical model of the FSM with the goal of understanding the behavior and working principles of the FSM. The AR4FSM application was envisioned to offer concept development and knowledge retention.

The development of a new AR application does not guarantee that its integration into the existing instruction methodology will be successful. The acceptance of the technology

is ensured if students are ready for the technology and it meets their expectations, as explained in the technology acceptance model (TAM) [22]. This model is used to perceive the ease of use and the usefulness of a technology, which can influence people's behavior toward accepting the technology. For engineering students, and, in particular, computer and software engineering students who are tech savvy, we can safely assume that they are ready to embrace AR technology as they already access online activities during lectures using mobile devices connected to internet through WiFi. Almost all students experience AR technology on or off campus in one way or another [11]. The application was tested with ECSE students to ascertain their expectancy and acceptance of AR. The students in ECSE have the capability to develop AR systems in one way or another. For example, electrical engineers are good at developing hardware, software engineers are better equipped with the ability to develop software applications, and computer system engineering combines the best of both worlds. The author, as well as the participants of this study, belongs to computer system engineering. We used a short questionnaire based on TAM3 to measure the rate of students' acceptance of the AR4FSM [23]. To our knowledge, no prior studies have reported the realization of a mobile application integrating AR technology to facilitate students' grasping of complex FSM concepts. It should be noted that, in this phase of the study, we did not focus on evaluating the effectiveness of AR4FSM in education.

This paper is organized as follows. Existing AR applications used in engineering education are presented in Section 2. Section 3 provides the details related to the development of the application and its features. The research study is described in Section 4. In Section 5, the results of the system evaluation are presented, followed by the discussion in Section 6. Finally, Section 7 concludes the paper.

## 2. AR in Engineering Education

Many existing studies in the broader literature examined the use of AR technologies in various educational disciplines and found that they have positive effects on students' motivation during the learning process [24]. AR has been used to explain basic educational concepts, such as the earth–sun relationship [25], and complex concepts in electromagnetism [19,26]. It is also used in numerous other fields, such as textiles [27,28] and training [29]. This technology-enhanced instruction methodology has been very effective in explaining difficult concepts to undergraduate students in the four basic STEM (science, technology, engineering, and mathematics) disciplines [7,16,30]. AR has also been widely used in various disciplines of engineering. For example, the CAM-ART application was designed for the teaching of building design, assembly projects in construction, and the understanding of the construction elements in the civil engineering discipline [31]. Engineers can now conduct virtual site visits and perform a comparison between the as-built and as-planned status of projects using AR applications [32].

A series of recent studies indicated that AR has been successfully deployed for the teaching of electrical and computer engineering concepts. For example, AR applications were deployed for the learning of electrical machine behavior by bridging the gap between theoretical explanations and laboratory practices [33]. A pedagogical virtual machine (PVM) proposed by [34] linked physical object activities with learning activities. The PVM collected data being transmitted to the embedded computer and processed them to produce a more meaningful representation. Based on the predefined learning design, these data were then translated to learning activities. The learner could visualize the workflow of the learning activities, track their progress, and obtain instant feedback based on their performance of the learning activity. AR-based applications for the assembly and exploration of modularized mobile robot tasks are already in use [35]. They provide assemblers with a more sophisticated learning experience and a deeper understanding of the software components inside embedded computing at the same time. AR applications were also used to enhance interaction with laboratory equipment, deepen conceptual understanding, and improve learner engagement [36]. This was achieved by integrating vision-based control with AR and touchscreen interaction. The mobile device augmented live video with

graphics when aimed at a laboratory testbed. The learners could manipulate it to control testbeds and perform experiments. In contrast to typical graphical interfaces, the devices were in charge of parts of laboratory testbed measurement, estimation, and control.

Most of the prior work related to the application of AR in the field of digital circuits and digital systems focused on either logic gates' functionality or the simple circuits constituted by these gates. For example, an interactive, marker-based AR application embedded the functionality of basic logic gates to provide a real-time laboratory environment [37]. An enhanced version provided the simulation of a logic circuit wired on a breadboard using the logic gates [38]. The application used image processing and pattern recognition to identify the ICs in pictures taken by the camera. Identified virtual objects, such as IC identifiers, pin configurations, and logic diagram information, were superimposed on the real-world image. The authors used the same technique to create an AR application that could determine the resistance values of resistors in a given circuit by scanning color codes and calculate theoretical current, voltage, and power.

As explained earlier, the FSM topic is important and tricky for students to understand. There are several simulation tools, such as ModelSim and WinState, which are used for functional verification of FSMs. ModelSim is a simulator in which hardware FSMs are described using VHDL, and it then shows the output as per the described FSM. Using the tool and interpreting the waveforms requires prior understanding of the basic concepts of FSM and proficiency in VHDL, which novice learners lack. Therefore, this method is good for applying the concepts learned but not suitable for teaching these concepts to beginners. WinState is a MS-Windows-compatible software tool used for tutorial-style teaching that aids the understanding of the mechanisms of an FSM [39]. It makes use of computer screen animations to visually convey the required design and analysis procedures, thus allowing students to construct the FSM, simulate it to verify the functionality, and detect errors. It requires the students to go through the complete design procedure before they can visualize the results. This lengthy process may be hectic and demotivating for the students. The aforementioned issues motivate the need for a tool that can convey complex FSM concepts in an easy, quick, and engaging way.

The student's addiction to mobile phones can be used to the student's advantage, and this was one of the sources of motivation for this research. This distraction due to non-academic use of mobile technology can be avoided by fostering its positive use for pedagogical purposes: to promote student engagement and create an interactive environment for meaningful learning [40]. This makes AR an ideal teaching support tool as it enables the visualization and manipulation of interactive models while encouraging students to participate in a more creative and fun process [41]. This motivated us to use mobile-based AR technology to enhance the instructions for delivering some fundamental concepts of digital system design. We transformed the concepts related to FSM into mobilized lessons, the term described by [42], transforming teacher-focused instructions into student-focused instructions. This was to help students gain long-lasting visual and conceptual knowledge and to gain an understanding of how students perceive and interact with classroom technology. The development of computational environments with more intuitive interfaces and a high level of interaction allows students to grasp concepts without much difficulty.

This paper addresses this gap by introducing the realization of a mobile-phone-based AR application for teaching FSM concepts, which are an integral component of digital systems, developed using successive approximation model I (SAM I) [43]. This model can enhance students' learning experiences and increase their understanding of complex FSM concepts by creating active learning experiences in a classroom setting and provides the students with a tool for independent learning outside of lecture hours. The students were provided with the FSM diagram drawn on paper. The AR4FSM superimposes the digital contents (an animation together with information extracted from the image) on a real-life image of an FSM and allows students to interact with it by changing the input and visualizing the transitions from one state to another through the movement of an

avatar. Furthermore, another important question that was addressed was how the students perceive this application.

## 3. Application Development

### 3.1. Finite-State Machine Model

Invented by Edward Forrest, the Moore finite-state machine is a mathematical model of computation used to simulate sequential logic and represent execution control flow [44]. This comes from "automata theory", which is a branch of computer science, and is used to model problems in many fields, including mathematics, artificial intelligence, games, or linguistics. The FSM has three different representations: a state table, a state diagram, and digital waveforms. A formal definition precisely defines the FSM and tends to avoid any uncertainties. A finite automaton is a 5-tuple $(Q, \Sigma, \Delta, \sigma, q0)$ where:

$Q$ is a set of states having a finite number of states;
$\Sigma$ is a set of symbols denoting the inputs;
$\Delta$ is a set of symbols denoting the outputs;
$\sigma$ is a transition function mapping $Q \times \Sigma$ to $Q \times \Delta$;
$q0 \in Q$ is a start (or initial) state.

A state transition diagram describing an FSM is shown in Figure 1, where each circular node represents a state, and each arc represents a transition labelled by "guard/action", where guards represent the input symbols triggering the transition, and action is the output symbol as a result of transition. The arc without a source state points to the initial state, i.e., state. In a single reaction, an FSM maps a present state $p \in Q$ and an input symbol $a \in \Sigma$ to a next state $q \in Q$ and output symbol $b \in \Delta$, where $\sigma(p,a) = (q,b)$. For any input sequence of words, a sequence of reactions generates a sequence of states and an output word [45]. The FSM below has two states, where $Q = \{\alpha,\beta\}$, $\Sigma = (a,b)$, $\Delta = \{\epsilon,u,v\}$, $q0 = \alpha$, and $\sigma : Q \times \Sigma \rightarrow Q \times \Delta$ is such that $\sigma(\alpha,a) = (\beta,v)$ and $\sigma(\beta,a) = (\alpha,u)$ with implicit self-transition $\sigma(\alpha,a) = (\alpha,\epsilon)$ and $\sigma(\beta,b) = (\beta,\epsilon)$.

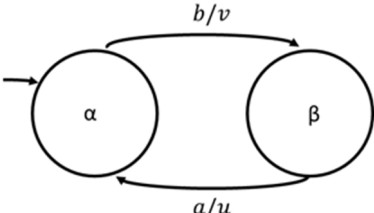

**Figure 1.** State transition diagram of a basic FSM.

### 3.2. Instruction Design Model

We used the successive approximation model (SAM 1) for instruction design, which is a simplified version of the ADDIE (analysis, design, development, implementation, and evaluation) model and can be applied to many learning situations. This is considered good for small projects and teams and suited our research, which required a working prototype earlier in the process. This model helps in building a quick prototype and makes it easier to reevaluate and assess the material. This is also particularly useful when feedback from multiple stakeholders needs to be collected with the aim of improving the quality of the product, leading to a more creative design.

### 3.3. Choosing AR Technologies

One of the key challenges in realizing AR systems is choosing the right hardware which meets the requirements. One option is to use wearable technology, such as head-mounted displays and smart glasses, but they suffer from limited battery life [46]. Another option is to use fixed-screen AR technologies, such as desktop computers, as a two-dimensional medium on monitor displays, but these displays cannot be used in traditional lecture

environments. Handheld devices, such as mobile phones and tablets, are compact and come with the required technology, and everyone knows how to use them [47]. An overwhelming majority of undergraduate students owns a smartphone, and most of them already bring them to the lecture environment [48]. For the aforementioned reasons, smartphones may be a preferred choice for realizing an AR system for use in lecture settings when compared with other devices. Furthermore, mobile devices are already equipped with a variety of sensors and cameras which can benefit AR [49].

A mobile-based AR experience can be realized through web services or a custom application [50]. A web-based application can access the AR systems using the internet, thus avoiding the hassle of installing the app on mobile and updating it in the future, as would be required if the app was not native to a particular system. However, web-based apps require an active internet connection all the time to run. In contrast to web apps, custom mobile apps need to be installed on the phone, but they can work offline without requiring an active internet connection. Furthermore, mobile apps are faster and more efficient, as all computations are performed locally on the device and no external communication over the internet is needed. Speed and the ability to work offline are two crucial factors that motivated the authors to design an AR system based on mobile devices [51].

### 3.4. Application Implementation

We developed an Android mobile application that is easy to deploy within classrooms. We chose Android Studio for developing our application due to its unified environment. It provides the capability to design a user interface, define functionality, and package the code into an *.apk* file, the standard extension for Android applications. We chose to work with API level 19 (kitkat) and above as it covers 95 percent of mobile phones that use Android (including entry level devices with small RAM) and enables access to more advanced APIs at the same time. Android's default programming language is Java, which we found to be advantageous due to its outstanding object-oriented programming (OOP) handling capabilities. Java also makes code modularity and maintenance much easier. Furthermore, code modularity and management becomes easy due to Java memory management attributes.

During the initial iterations, the OpenCV library was used for object recognition. In the FSM feature, a marker-based AR system was used. The printed FSMs on paper consisted of a marker in each of the states to make the application aware of their positions. To recognize these markers, OpenCV employs machine learning to create a model for detection and then performs the AR algorithm over the FSM model. Although machine training produces relatively high accuracy, the performance in terms of speed of detection is not good or robust enough to undermine user satisfaction. The initial version of the application could handle only a maximum of four input conditions and two states at the most.

In the final iteration, we made the transition from the not-so-robust, marker-based algorithm to a text-based detection algorithm. The API used for this was Google Mobile Vision, which has the best text-detection feature currently available by far, thus making the FSM simulation much more robust. It was also decided that two states were not enough to help students learn concepts fully, but increasing the number of states raised the level of difficulty and made it hard to build a working prototype. This required the remodeling of the framework of the code in order for it to have an automated method for dealing with any number of states in an FSM. In practice, however, it was ideal for the app to have a maximum number of four states in the FSM due to the capabilities and limitations of the detection API being used. This was a major change from the previous iteration, where the application was designed for two states only. This means that, with further development of the API, the code provided is easily modifiable to handle more than four states. The instruction materials for two-, three-, and four-state FSMs are shown in Figure 2a, Figure 2b, and Figure 2c, respectively.

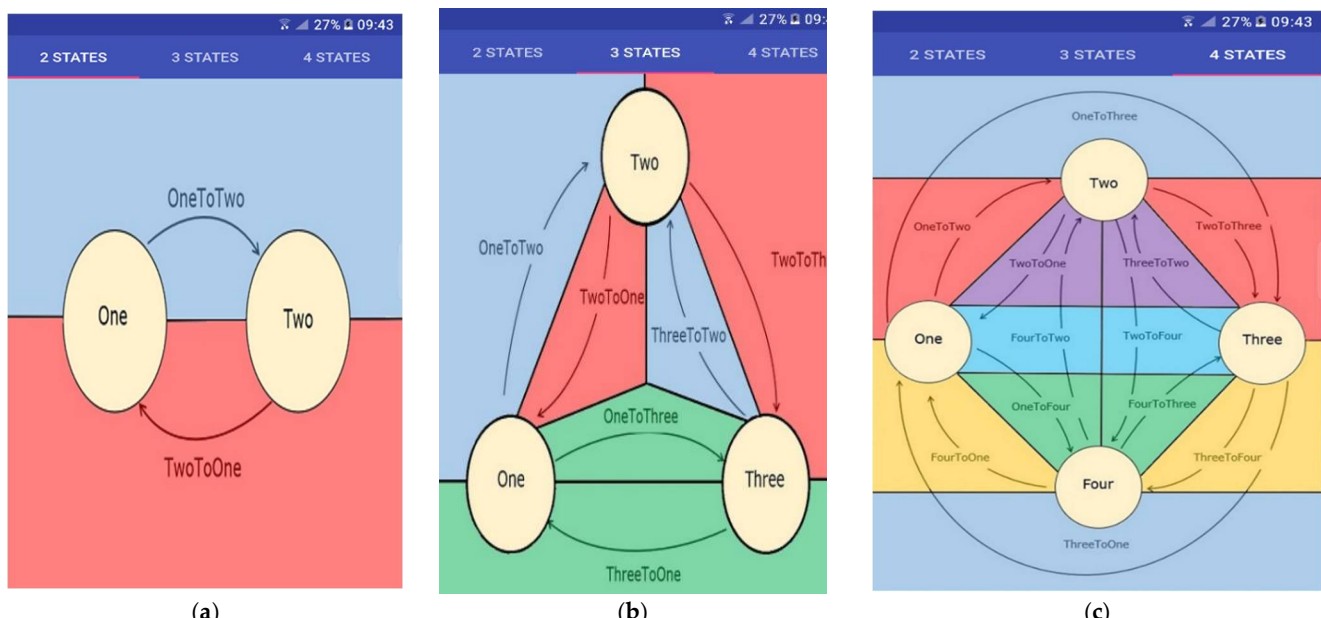

**Figure 2.** Instruction material for FSM with (**a**) 2 states (**b**), 3 states, and (**c**) 4 states.

*3.5. AR4FSM Software Architecture*

The software architecture of the AR4FSM is outlined below in Figure 3. The AR application was based on markerless tracking technology that targets portable devices and smartphones in particular. It uses image recognition and, more specifically, recognizes the text using Google Mobile Vision APIs. The screen controller includes placement of input boxes and switching screens within the app, while the camera controller is used to validate camera permissions and open a camera view within the app. Once the user enters the camera view, the interaction with real-world objects is defined by the tracking methods that are used. Within the Android studio, we defined activities that describe the actions that different user inputs correspond to. For each action within a feature, an individual activity was defined for it, and a main activity was also designed for facilitating overall interactions. Additionally, we set a layout for each feature that guides the user through different views in the application as seen from the user interface. The functionality of the overall app was defined separately into multiple classes for code reusability, wherein each feature has its own specific processing algorithm that it undertakes on detecting the object it is looking for.

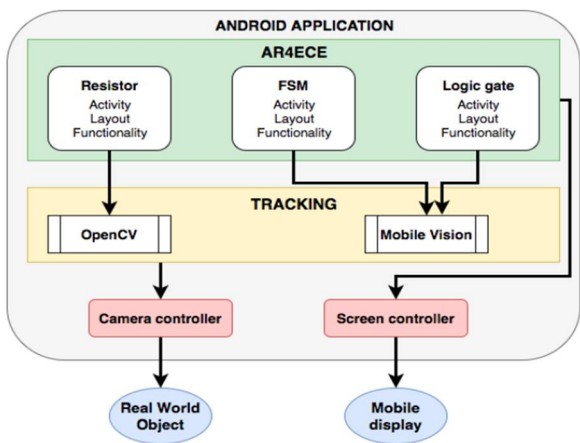

**Figure 3.** Software architecture.

The front-end code structure within the "res" folder is shown in Figure 4a. When packaged, all the files have a separate configuration process, which is dependent on the type of file, that is followed when the app is built. Therefore, certain file types (i.e., as distinguished by front-end and back-end code) need to be stored in the correct folders. The subfolders that were modified during the front-end development for AR4FSM are shown in extended format to the right as well. There are only definitions of the subfolders and files that were altered or edited during the developmental process which may be required for future developers in case the app's functionality is extended.

- anim—It is a packed animation. This is an animation natively created by Android widgets;
- drawable—The miscellaneous images, icons, and figures that are used in the application;
- layout—This contains the descriptions for the page layouts. In essence, every activity describes the user views of the app and the interactions that change the view of the app within every feature;
- raw—The audio files or any haar cascade file for OpenCV's object recognition should be stored here;
- values—This is used for defining constant values for the attributes in the application such as color, text, and other fixed values that are used across the whole application.

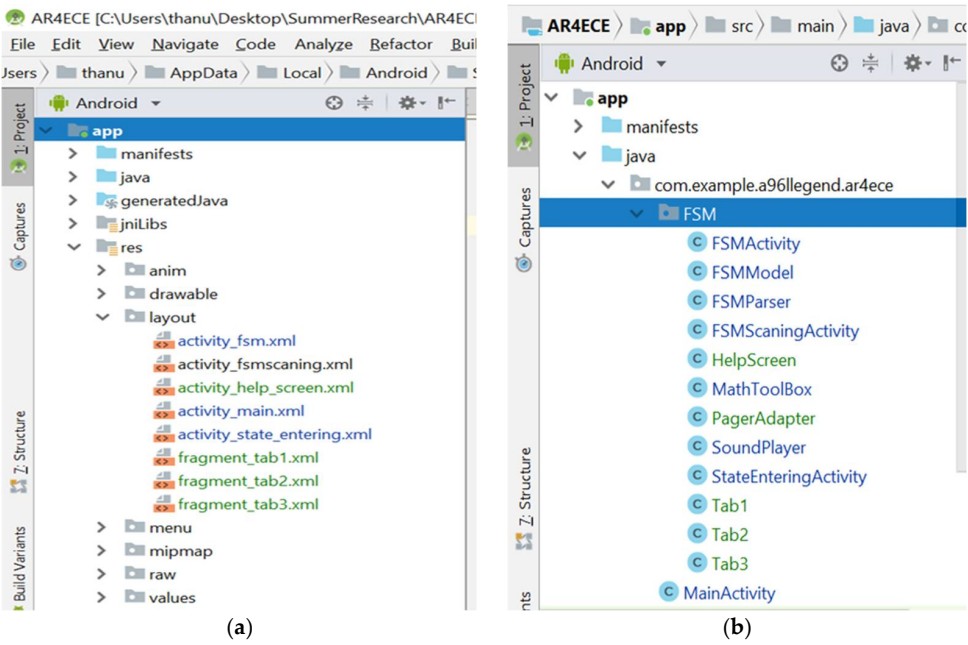

**Figure 4.** Code structure: (**a**) front-end, (**b**) back-end Java classes.

The back-end functionality of the application was implemented in Java with different classes, as shown in Figure 4b, with "MainActivity" being the most important class. It defines the logic for switching from the homepage view to a particular feature's view. Here, we define the classes that were adopted for back-end development:

- FSMActivity—Manages text detection and captures the user's input while running the FSM feature;
- FSMModel—This contains the model of the FSM which defines the structure and framework of FSMs;
- FSMParser—This generates Boolean-type equations from the text detected on scanning an FSM. These equations correspond to the transactions that the FSM needs to make. It builds upon the scanned result obtained from FSMScanningActivity;
- FSMScaningActivity—Manages text detection when the app is scanning the FSM from the paper;
- HelpScreen—The view provided to the user, as shown in Figure 4;

- MathToolBox—Calculates the path the stick man animation should take between two fixed points;
- PagerAdapter—Allows for switching of tabs in the help screen;
- SoundPlayer—Customized sound player for playing audio during animation;
- StateEnteringActivity—The view provided to the user for entering state names;
- Tab1—Help screen guide for FSM with two states, as shown in Figure 2a;
- Tab2—Help screen guide for FSM with three states, as shown in Figure 2b;
- Tab3—Help screen guide for FSM with four states, as shown in Figure 2c.

### 3.6. Tracking

In many AR applications, digital world contents are overlaid on real-world contents, necessitating the device's ability to perceive the surroundings and the user's movement in real time. This phenomenon of recognizing an object or scene is called tracking. There are three methods of tracking: marker-based, markerless, and hybrid tracking [52]. The marker-based AR approach augments digital information by recognizing objects and locations through the camera in a smartphone by means of markers associated with the objects to be recognized. These markers cannot be identified by other sensors such as digital compasses or GPS, which is a downside of marker-based AR. This technique suffers from the drawback that it requires the positioning of the markers on the object, which should be visible and cannot be obscured by any other objects throughout the process of augmentation. To supplement present, marker-based AR challenges, Park and Park [53] investigated invisible-marker-based AR, such as that which employs infrared markers. Vision-based AR technologies augment digital information by detecting the attributes of objects and tracking them after they have been recognized. These approaches offer the advantage of being able to follow objects without the usage of a marker, but they are challenging to implement because of the need for real-time image processing and augmentation.

### 3.7. Application Features

The AR4FSM is a task-based application where students are required to complete a task by interacting with the provided instruction material and learning the concepts through a sense of immersion. The students are provided with a manual finite-state machine diagram with all the states and the transition conditions as shown in Figure 5 below. When first turning on the app, the user is brought to the homepage, where they are prompted to click on the FSM option on the screen to begin the application, as shown on left of Figure 5a. The initial version of the application also included features such as gate logic and resistor value calculation using AR, which are not the subject of this paper and were, therefore, excluded from the final version. The user is then prompted to enter the state names of their FSM into the required fields, as shown on right of Figure 5a. Complex FSMs need a key in the diagram set by the user to differentiate between states and conditions. Therefore, to maximize ease of use, state names are entered beforehand. This is opposed to having the user manually enter a key for each state when designing the state diagram, which would be more tedious and time consuming. The names of the states entered should precisely match the names of the states present in the state diagram, and the names are case sensitive as well. There is also an option to change the avatar that is used in the FSM animations, as shown in Figure 5b.

Following the entry of the state names, the user is instructed to scan the FSM by pointing the camera at the FSM sketched on the accompanying handout shown in Figure 6a. As illustrated in Figure 6b, once the scan is complete, the screen is populated with FSM parameters such as state names, transition conditions, inputs, outputs, and their initial values. The user is next asked to confirm if the information obtained from the scan is accurate. Upon student confirmation, the user can select "Continue" to continue if they wish. The user has the option of scanning the information again if it is wrong or missing. The FSM's transition conditions are assigned automatically based on the diagram's position and number of states.

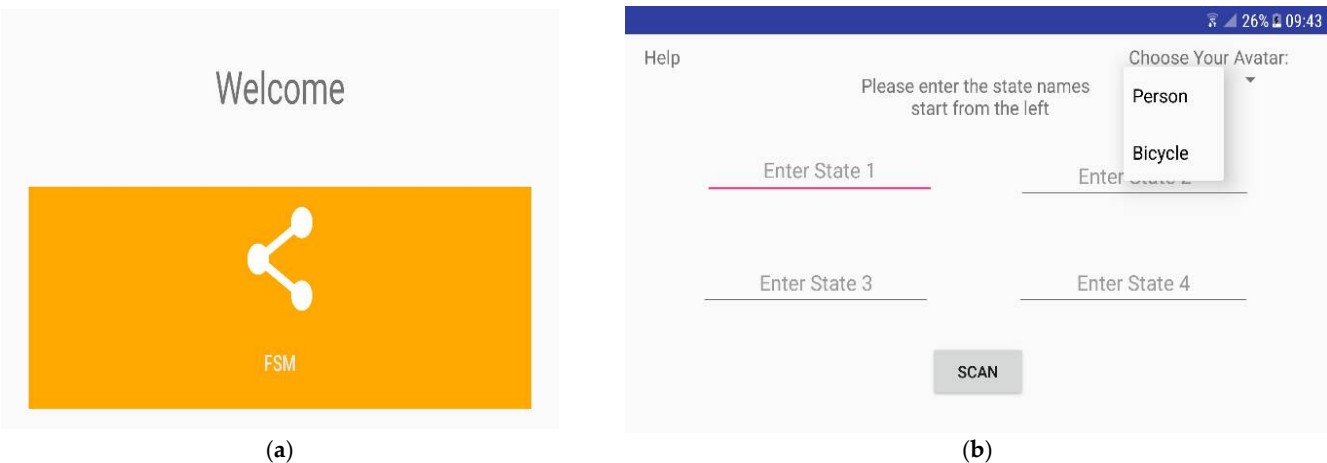

**Figure 5.** Screenshots: (**a**) welcome screen, (**b**) entering state names and selecting avatar.

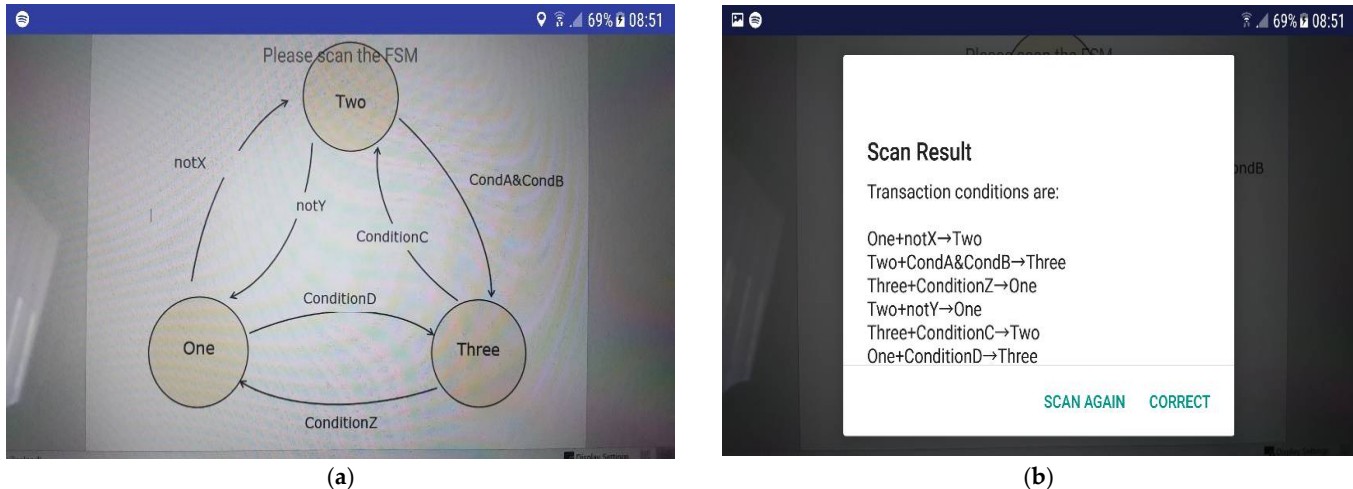

**Figure 6.** Digitalization of the object: (**a**) instruction material as object, (**b**) scan results.

A help button is provided to access the help menu, as shown in top-left corner of the Figure 7a. As you enter the help screen, a pop-up dialogue box appears immediately, providing tips on the design of the FSM, as shown in Figure 7b. Instructions are also provided on how to concatenate and invert conditions. To concatenate the conditions, the "&" symbol is used in between and is then split into its constituent parts during the simulation. The "and" keyword was originally an option as well but was removed in the final iteration as it created many unwanted results due to the word "and" being contained in many other names. In order to invert the conditions, the "not" keyword was used at the front of the condition, for example, "notConditionX". Alternate expressions used for not conditions, such as "!" and "~", were disregarded as the text recognition had difficulty recognizing them, and smaller symbols were often misinterpreted.

The last step for the user is to run the actual visual simulation. Their avatar, an animated character, as mentioned earlier, appears on top of the start state. This avatar can be changed by the user through the option provided on the screen. The user can now simulate the FSM and visualize the transition by interacting with it by changing the input values. The list of input signals and their initial values are placed at the bottom of the screen, and the user can toggle the values between "0" and "1" by simply tapping on the corresponding input name. After setting the desired input values, the user taps the Next button, thus allowing the avatar to make a transition to the next state based on the evaluation of transition conditions. The avatar, present in the current state, as

shown in Figure 8a, moves to the next state along the path of true condition by generating pleasant music.

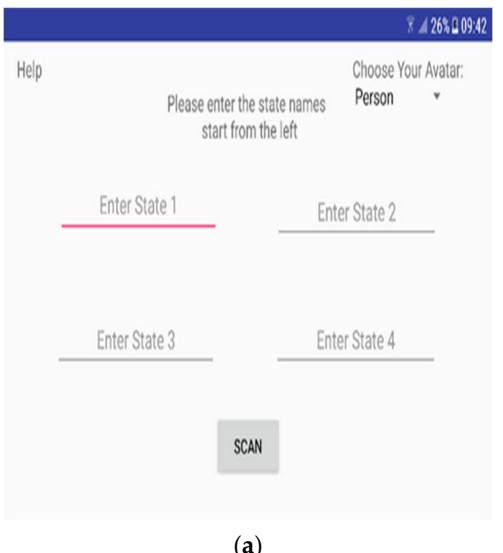 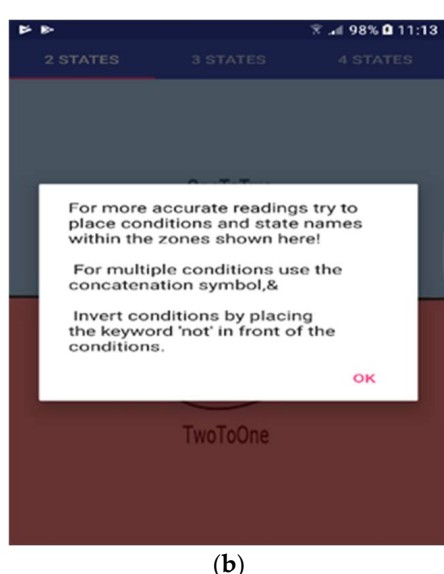

| (a) | (b) |

**Figure 7.** Help screen: (**a**) screenshot showing help button, (**b**) help screen dialogue box.

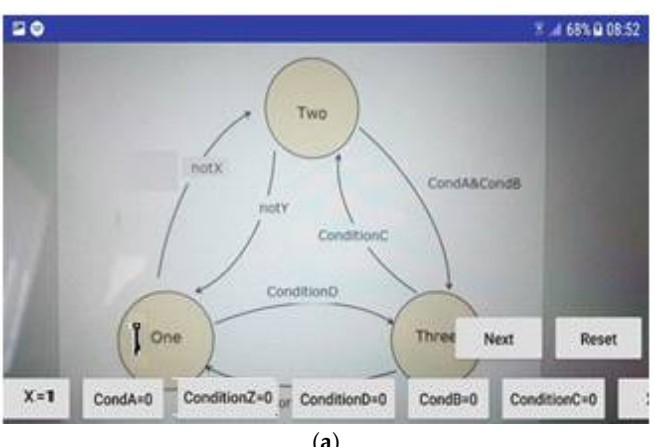 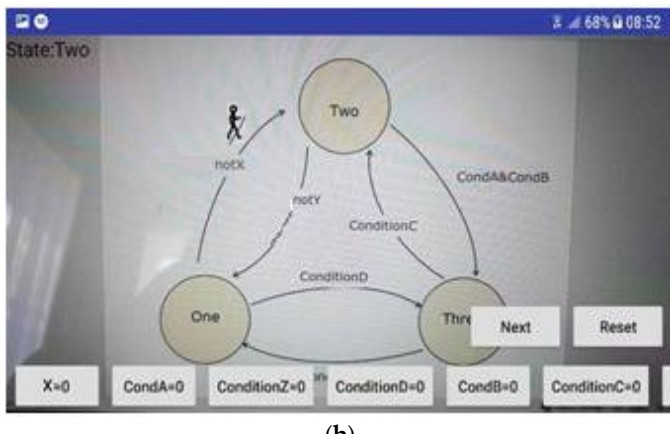

| (a) | (b) |

**Figure 8.** (**a**) Visual simulations of an FSM, (**b**) transition from state "One" to "Two".

The avatar traverses from the current state to the next state, i.e., from state "One" (when X = 1) to state "Two", when condition "notX" is true (X = 0), as shown in Figure 8b. Once it reaches the destination state, distinctive music is generated. If the conditions are not met for any state transitions, then, when Next is clicked, there is a short animation of a figure moving around and then back to the same state.

A button provides a reset option, which returns the animation to its initial state. When there are no conditions in the FSM, the animation progresses from one state to the next in the order entered by the user each time the Next button is pushed. The avatar's transition from state "One" to state "Two" is shown in Figure 9a, and from state "Three" to "One" is shown in Figure 9b. When the avatar moves to the next stage after evaluating inputs, the name of the destination state is displayed on the screen to make it more user friendly and easier to follow.

The FSM state diagram becomes more packed and cramped with text as the number of states increases, as shown in Figure 10. Due to the shrinking size of the zones that each condition is supposed to fit in, diagrams with five or more states become impossible for users to design and test effectively. Following extensive testing with a variety of other students, it was determined that, for best performance, AR4FSM should be confined to four

states and a maximum of 15 input conditions as this gives the best accuracy. Extending the number of states requires only modifying a few variables in the software.

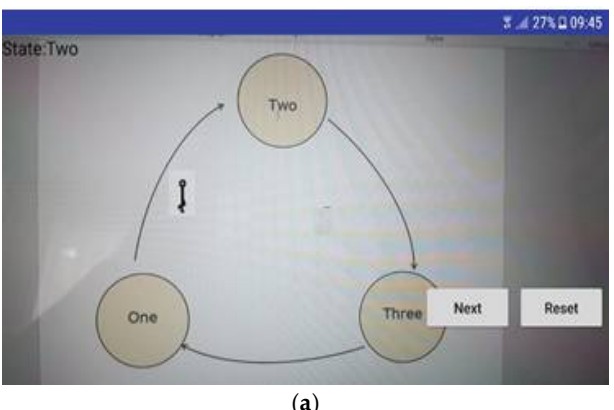 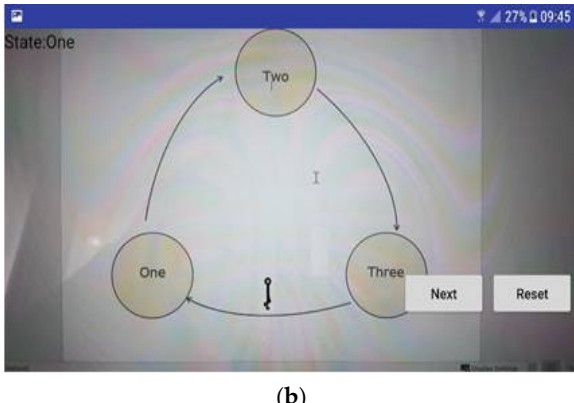

(**a**)        (**b**)

**Figure 9.** State transitions with no conditions: (**a**) state "One" to state "Two", (**b**) state "Three" to state "One".

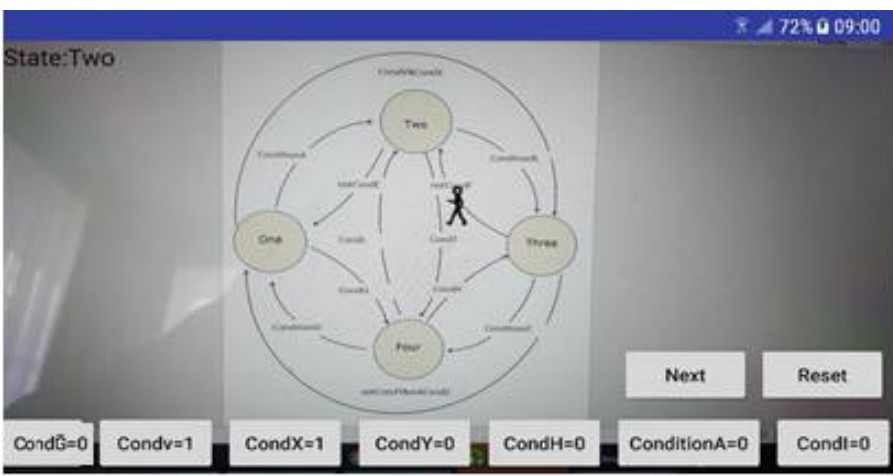

**Figure 10.** State transitions diagram with 4 states cramped with text.

## 4. Materials and Methods

### 4.1. Research Objective

The objective of this study was to improve the existing method of instruction by incorporating modern technologies to facilitate the delivery of FSM principles while also engaging and motivating students. The major component was to design and develop an application that allows them to interact with it and perform visual simulations. The second goal of this study was to measure the rate of students' expectancy and acceptance of this application being deployed in class in the future to enhance engagement, motivation, and understanding.

### 4.2. Research Subject

A total of 60 students (aged 19–22) from the department of Electrical, Computer, and Software Engineering (ECSE) took part in the study. Both male and female undergraduate students were included. All the participants were engineering students, with the bulk of them enrolled in computer and electrical engineering programs in their second and third years. Furthermore, the participants had previously encountered AR of some kind. The participants were given a smartphone with an AR app installed on it. Alternately, students could download the AR4FSM from the app store onto their own phone if they wished to do so. Students were given a task to complete in order to evaluate the app's features, such as bit stream pattern identification using FSM. The instruction material comprised a printed

handout that contained a three-state FSM for recognizing a bit stream with a length of two bits similar to the one shown in Figure 2. Students were given a short demonstration prior to the experiment and went through the whole process of scanning the FSM and changing the input, resulting in state transition which could be visualized through text, avatar movement, and music. Different sounds are generated when the avatar is making a transition from one state to another state and when it reaches the destination state. This procedure is similar to the one described in Section 3.6, which explains the application features. Once a participant had experienced the AR4FSM by accomplishing the task of traversing different states of the FSM, they were asked to fill out a paper-based questionnaire.

### 4.3. Research Sample Selection

All ECSE students who were potential participants received an invitation to participate in the study via the department mailing list, along with a participation information sheet. By filling out a consent form and returning it to the researchers, the students gave their consent to participate in the study. The researchers then called these participants and set up a time to conduct the survey. First, 60 students who gave their consent to participate in the study were contacted. The next participant was contacted if a person wanted to withdraw after obtaining consent. The facilitators and students who took part were from the same department. The study was conducted at the University of Auckland City Campus during university hours but not during lecture time. Participants were selected on a first-come, first-served basis.

### 4.4. Apparatus

Participants could interact with the real-world FSM drawn on the handout using a smartphone and control the movement of the avatar by changing the inputs in the AR environment. The study was mostly conducted with an Android-based Samsung Galaxy S10 Wi-Fi SM-T700 16 GB model running Android 9.0 Pie with Samsung Exynos Octa-Core CPU processors, $2 \times 2.73$ GHz Mongoose M4 and $2 \times 2.31$ GHz Cortex-A75 and $4 \times 1.95$ GHz Cortex-A55, an ARM Mali-G76 MP12 GPU graphics card, and 3 GB LPDDR3 RAM. Alternatively, students could download this application from the Play Store and install it on their phone. The code was also available from a GitHub repository which could be accessed by contacting the corresponding author. We made use of the built-in technologies of the mobile for AR system, such as the camera to capture real-world views, a touch screen for interaction, and speakers to play music. Unlike many other existing applications, the instruction contents were not fixed, and any FSM drawn on a paper following the guidelines provided in the help menu could be used as an instruction material.

### 4.5. Questionnaire

A short questionnaire comprising two parts was designed to evaluate the mobile AR-based application The first part used a 5-point scale, ranging from strongly disagree to strongly agree, to capture the response of participants, and the second part collected open-ended feedback in terms of likes/dislikes and suggestions. Six questions in the questionnaire focused on the quality of the application design and how easy it was to use the application, as well as the learning experience of the FSM, including the learning interest, engagement, active learning, level of understanding, academic outcome, and the extent to which the participants would like to have the respective learning tool applied in their class. As mentioned earlier, we based our questionnaire on TAM3 [54] to measure the rate of students' acceptance of the use the AR4FSM app. We selected and adopted questions presented by [31]. We omitted simple technology-related questions as these were not applicable to engineering students, especially ECSE students who are tech savvy and highly experienced. The questionnaire enlisted the questions given in Table 1.

**Table 1.** Survey questions.

| Sr. # | Question |
|---|---|
| Question 1 | The application was easy to use (expectancy). |
| Question 2 | Application can help in delivery of the contents in an easier way (expectancy). |
| Question 3 | Application can make the course contents more engaging (acceptance). |
| Question 4 | Application can help in better understanding of the FSM concepts (acceptance). |
| Question 5 | Application can help in achieving course learning outcome (acceptance). |
| Question 6 | AR application should be used in a classroom environment acceptance). |

## 5. Results

A total of 60 students participated in the study: 46 (76.67%) male and 14 (23.33%) female students. All of them were undergraduate students from the ECSE department. The findings of the evaluation of the application are depicted in the graph shown in Figure 11, which shows that 81.67% (mean = 3.96, SD = 0.96) of the students agreed that the mobile app was easy to use, of which 26.67% strongly agreed that app was user friendly. In response to question 2, 85% (mean = 42; SD = 1.04) of the participants agreed that the application delivered the FSM-related contents in an easier way. Similarly, more than 88.33% (mean = 4.4; SD = 0.84) of the students agreed that AR made the lecture contents more engaging, as evident from the response to question 3, which related to engagement. In response to question 4, 90% (mean = 4.4; SD = 0.84) of students agreed that the use of the AR application helped them to better understand the concepts, and 86.66% (mean = 4.2; SD = 0.94) of students agreed that it helped in achieving the learning outcomes. Only 70% (mean = 3.6; SD = 1.09) were in favor of the use of AR technology in the class.

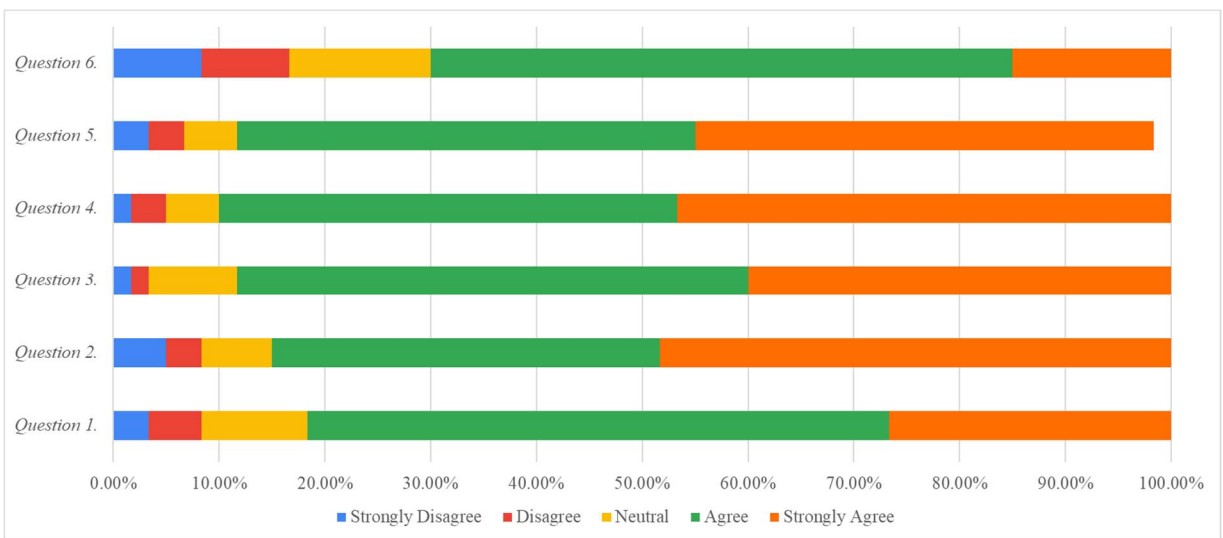

**Figure 11.** Responses to the survey questions.

In response to an open-ended question related to anything which a user might like or dislike in this app, the positive comments were:

- The animation character and sound did draw my attention;
- The application did engage me and provided me with immersive experience;
- It was wonderful and exciting way of learning;
- It is helpful when checking the correctness of the designed FSM without the need of coding it.

Students disliked the following:

- It as bit on slower side;
- Avatar is not attractive and could be better than it is;
- It can eat up a lot of lecture time;

- It needs multiple scans if FSM is not drawn clearly on the paper.

Most significant suggestion for improvement given by the students were:

- A running commentary or more explanation of the process along with the music;
- Add more Avatars which are rich in colours;
- This app should be used during the sophomore year when students are first introduced to finite state machine concepts.

## 6. Discussion

The findings show that the mobile AR application, AR4FSM, received positive feedback from the students with regard to the ease of use. One of the most crucial factors in designing an application is its ease of use, as it plays a pivotal role in the success of the application. According to researchers, inadequate AR application design in terms of usability considerations may cause distractions and reduce students' overall learning effectiveness. An application that is not easy to use often results in users abandoning it. Success lies in making it easier for the user to learn the features and providing a greater user experience. This is in line with the findings of [52], who discovered that the ease-of-use factor can reduce the level of satisfaction among students, and, as a result, the researchers advised that ease of use should be improved and strengthened.

The students overwhelmingly agreed on the point that AR4FSM made the delivery of content easier. This is largely because AR technology can present information in a way that cannot be otherwise visualized by students' minds in the classroom. This AR app provided the students with extra digital information about FSMs, which helped in explaining abstract and difficult contents that would not be easy to understand otherwise. Furthermore, adding extra information in the form of a visual model not only helped students to get a deeper understanding of the lesson topic but also entrapped their attention and motivated them to study. AR is especially good for visual learners as information is offered to them in their preferred style.

Similarly, the students were of the view that AR made the lecture content more engaging. This can be attributed to the involvement of emotion when using AR, which is the key to boosting student engagement. Through AR, students become part of the lesson, and they become emotionally attached to the topic, which is not possible with the use of text only. For any FSM drawn on a paper, based on the current state and the status of inputs, the student can make a prediction about the next state using his knowledge, but there is no way to see it happening and get feedback. When using AR4FSM, students can change inputs, then visualize the transition of the avatar from current state to predicted state. The avatar's movement from one state to another based on input selection gives immediate feedback by confirming their input option. If the avatar proceeds to the projected next stage along the predicted path, it affirms the student's understanding, which has a good impact on student's motivation. There is no need to mention that AR is a modern technology that immediately grasps the attention of students and that control over learning provides the confidence. All these factors contribute to motivating the student [55]. According to the ARCS model of motivation, the combination of challenge and feedback in the form of verifying their performance in learning activities enhances confidence and satisfaction [56].

The findings show that when students were asked about their understanding of FSM-related concepts, the AR4FSM received positive feedback. This is in accordance with the earlier studies which verified that most engineering students prefer learning through visual resources and learn better in an interactive learning environment [57]. The use of an AR-based, interactive pedagogical tool in engineering education enhances learning quality. The students agreed that AR4FSM could aid in the achievement of learning objectives because it presents the concepts related to FSM in a simple and meaningful way that looks like real life. Students can visualize and examine the impact of stimuli in an interactive way. With the help of technology, they experience something that might not be possible otherwise. Though most students agreed to the use of AR4FSM inside the classroom, the result was slightly on the lower side. This could be due to the amount of time spent on downloading

and installing the application, device limitations, and connectivity. The prototype version might also be the reason for the slightly lower level of acceptance. Additionally, the use of AR in the engineering classroom eats up a lot of time, which is not appreciated by engineering students as it forces the lecture to cover the remaining contents at a slightly quicker pace. This is also reflected in the comments made by the students in response to the open-ended questions. However, AR4FSM does not impose temporal and spatial boundaries, meaning that it can be used inside as well outside the lecture hours. This type of blended learning is already gaining popularity as restrictions due to the pandemic start easing, and universities are already migrating to this pedagogical style where face-to-face lectures are being blended with online activities or it is seen as an alternative to face-to-face lectures. This integration of AR technology with traditional or instructor-led teaching allows the students to enjoy the best of both worlds and caters to the needs of all types of learners. The instructor-led portion of teaching allows learners to engage easily, while the online portion allows for the management of the pace of learning [58]. This means students can learn while away from class or in distance education settings.

Another point raised in the open-ended discussion was about the display size; it could hinder the engagement of a student if the device has a smaller display which makes it hard to read the text. Gabbard and Swan II [59] came to similar conclusions, stating that a small display could be an issue if AR is too complicated for existing mobile devices and the interface has too many elements and menus to manage. Furthermore, the effectiveness of the application depends on the quality of interaction. Any hindrance to interaction results in hindering the learning process. This issue was already addressed in this research by the restriction of the number of FSM states under investigation. This problem could also be solved using tablets, which have bigger displays compared to mobiles, but this is the only benefit they offer. Tablets are a less portable version of smartphones, and they are less commonly used for messaging and calling. Smartphones, on the other hand, offer all those functionalities offered by the tablet, and it is much easier to carry them. Another option could be the use of fixed-screen AR technologies, such as desktop computers, as a two-dimensional medium on monitor displays, but these displays cannot be used in traditional lecture environments and can only be used either in lab setups or small classrooms where computers are available to all students.

### 6.1. Implications of the Study

The findings of this study demonstrate that AR4FSM creates opportunities for teachers to present abstract concepts in a way that students can grasp. The interaction offered by AR can help to enhance the classroom experience and inspire the minds of students. One of the major issues in adopting the AR technology in teaching is its seamless integration into instruction methodology, which is primarily hindered by the lack of teacher and student ability to adapt to new technology. This mainly depends on the ease of use of the technology and the students' belief that the use of the technology will help improve their academic performance. Therefore, it is important to assess the acceptance of applications before using them in large classrooms, such as at the University of Auckland, where the class size exceeds two hundred students. The outcome of this study will help researchers to make AR technology more acceptable for the students and integrate it into instruction methodology in a meaningful way.

### 6.2. Limitations of the Study

This application is only available for Android operating systems running on Android version 8.1 and above. The questionnaire used could have been more detailed. In addition, the results were based on a self-report study, as it involved participants filling in questionnaires regarding their user experience of the AR4FSM app. A qualitative method with detailed interviews and observation of the participants could have been more prolific and factual. The participants were recruited using a convenience sampling technique, as it was

fast and easier, thus the study is subjected to limitations in generalization and inference, resulting in low external validity of the research.

## 7. Conclusions

In this study, a smartphone-based AR application called AR4FSM was developed with the goal of making complicated FSM principles more understandable and interesting for students. Students responded positively to this application, and we recommend using it in the classroom for teaching FSM principles with some improvements to supplement instructional materials such as the handouts. It is an excellent learning tool since it allows for the understanding of concepts gained through deeper interactions with the real world. In addition, this strategy is better suited to some students' learning styles, resulting in a more comprehensive teaching approach that more successfully responds to all students' needs. If we provide information to the mini millennials using this sort of technology instead of standard lectures, they are more likely to be engaged with the learning process.

In the future, first of all, we plan to update the AR4FSM application based on the feedback received from participants and reviewers to further improve its acceptability in the classroom setting. This will be performed by redesigning the engineering instruction method related to the FSM by embedding the AR4FSM experience into existing instruction material in a meaningful way. We also intend to find the extent to which it motivates the learners and identify if students' learning outcomes are improved as a result of its use.

**Author Contributions:** All authors contributed extensively to the work presented in this paper. Conceptualization, M.N.; methodology, M.N., M.L. and J.C.; software, M.L. and J.C.; validation, M.L., J.C. and M.N.; formal analysis, M.N., M.S.; investigation, M.N., M.S.; resources, M.N.; data curation, M.N., M.L. and J.C.; writing—original draft preparation, M.N., J.C. and M.N.; writing—review and editing, M.N. and M.S.; visualization, M.N. and M.S.; supervision, M.N.; project administration, M.N.; funding acquisition, none. All authors have read and agreed to the published version of the manuscript.

**Funding:** This research received no external funding.

**Institutional Review Board Statement:** This study was approved by The University of Auckland Human Participants Ethics Committee (reference 022118).

**Informed Consent Statement:** Informed consent was obtained from all subjects involved in the study.

**Data Availability Statement:** The datasets used and/or analyzed in the current study are available from the corresponding author on reasonable request.

**Conflicts of Interest:** The authors declare no conflict of interest.

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
