# Peer review of "AR4FSM: Mobile Augmented Reality Application in Engineering Education for Finite-State Machine Understanding"

_education, doi:10.3390/educsci12080555_

Round 1
Reviewer 1 Report
The study shows the development of an interesting application based on augmented reality to teach and learn the concept of finite state machine. The authors are right to propose a blending between technology and the professor, and not the technology to replace the teacher. I consider that the study has a great potential to enhance learning in engineering; however, there are some minor and major issues that prevent its publication in its current form.
Minor issues
· Define terms the first time you use them, and then use the acronyms (e.g., AR, ECSE).
· Arrange the keywords in alphabetical order.
· Unify the terms to refer to the developed application. The author(s) use the terms AR4FSM, AR4FS, and AR-FSM indistinctively.
· The introduction section contains more information than necessary. Subsections 1.1., 1.2, and 1.3 should focus on briefly describing the technology or the concept to be explained and use external references to guide the readers on what to read if they want to know more about the topics.
· I consider that Figures 1 and 3 are not necessary.
· Table 2 is not necessary. The displayed information could be easily explained in the text.
· When presenting the results, the author(s) should use the exact values. Try not to use vague statements such as “Around 81.67% of students,” “around 85% of the participants,” “more than 88.33%,” “Only ~70%.”.
· There was an apparent unconscious error in the first paragraph of the discussion section.
· Most of the references are relatively old. The author(s) are invited to update the used references.
Major issues
· The purpose of the study is to facilitate the delivery of FSM principles while also engaging and motivating students. With respect to that purpose, the author(s) stated two research results:
1. The findings show that when students were asked about their understanding of FSM-545 related concepts, the AR4FSM received positive feedback.
2. The tool engaged and motivated the students.
However, the implemented survey does not allow identifying if students actually understood the concept. I recommend that the author(s) conduct a pre-test/post-test with control study to identify if students learning outcomes improve when using the application.
Additionally, the survey does not allow identifying if the extent of motivation of the students. To do so, I recommend that the author(s) implement any motivation survey such as the IMMS de Keller (2010).
· The authors failed to state the implications of the study, its limitations, and possible directions for future research.
· Review the manuscript in terms of English writing.
Author Response
Response to Reviewer 1 Comments
Thank you very much. We appreciate the time and effort taken to provide feedback on our manuscript. We are grateful for the insightful comments and valuable suggestions for improving the article. Following are the responses to reviewer’s comments.
Minor issues
Point 1: Define terms the first time you use them, and then use the acronyms (e.g., AR, ECSE).
Response 1: Thank you very much for pointing it out. We agree with the reviewer and necessary changes have been made.
Point 2: Arrange the keywords in alphabetical order.
Response 2: We have arranged the keywords in alphabetical order.
Point 3: Unify the terms to refer to the developed application. The author(s) use the terms AR4FSM, AR4FS, and AR-FSM indistinctively.
Response 3: We have made corrections and we are using only one term, AR4FSM, to refer to the developed application.
Point 4: The introduction section contains more information than necessary. Subsections 1.1., 1.2, and 1.3 should focus on briefly describing the technology or the concept to be explained and use external references to guide the readers on what to read if they want to know more about the topics.
Response 4: We agree with the reviewer. We have moved subsection 1.1 to the Application development section where technical aspects are discussed. We think that this discussion fits well in this section. We have revised subsection 1.2 and excluded unnecessary details. Some new discussion is also added in this subsection as suggested by reviewers. However, we think that subsection 1.3 is important as it provides necessary background information which makes it easier for the reader to follow the discussion in later sections.
Point 5: I consider that Figures 1 and 3 are not necessary.
Response 5: We agree with the reviewer’s comment and we have removed Figures 1 and 3 in the revised manuscript.
Point 6: Table 2 is not necessary. The displayed information could be easily explained in the text.
Response 6: We agree with the reviewer’s comment and we have removed Table 2 in the revised manuscript.
Point 7: When presenting the results, the author(s) should use the exact values. Try not to use vague statements such as “Around 81.67% of students,” “around 85% of the participants,” “more than 88.33%,” “Only ~70%.”.
Response 7: We agree with the reviewer’s comment. We have provided exact percentages in the revised manuscript.
Point 8: There was an apparent unconscious error in the first paragraph of the discussion section.
Response 8: Sorry for the unnecessary sentence. We deleted this sentence.
Point 9: Most of the references are relatively old. The author(s) are invited to update the used references.
Response 9: Thank you very much for pointing it out. We have updated the references in revised the manuscript.
Major issues
The purpose of the study is to facilitate the delivery of FSM principles while also engaging and motivating students. With respect to that purpose, the author(s) stated two research results:
- The findings show that when students were asked about their understanding of FSM-545 related concepts, the AR4FSM received positive feedback.
- The tool engaged and motivated the students.
However, the implemented survey does not allow identifying if students actually understood the concept.
Point 1: I recommend that the author(s) conduct a pre-test/post-test with a control study to identify if students learning outcomes improve when using the application.
Response 1: We appreciate the reviewer’s suggestion but it is beyond the scope of this study. The main focus of this study is to develop the AR application and find student expectancy and acceptance. In this phase of the study, we are not focusing on evaluating the effectiveness of AR4FSM in education.
Point 2: Additionally, the survey does not allow identifying if the extent of motivation of the students. To do so, I recommend that the author(s) implement any motivation survey such as the IMMS de Keller (2010).
Response 2: We appreciate the reviewer’s suggestion but it is beyond the scope of this study. In this phase of the study, we were not sure if this technology can be deployed successfully. In the future, we intend to update the application based on the feedback from participants and use it in the classroom setting where all the students will go through a similar experience. We intend to evaluate the effectiveness and extent of motivation. However, thanks for suggesting the IMMS, and we will definitely consider implementing it when finding the extent of motivation in the future.
Point 3: The authors failed to state the implications of the study, its limitations, and possible directions for future research.
Response 3: Thank you very much for pointing it out. We have included this information in the revised manuscript.
Point 4: Review the manuscript in terms of English writing.
Response 4: We sincerely appreciate the reviewer’s comments. We have polished this
Manuscript, if the reviewer is still not satisfied, we can avail of the proofreading services.
Author Response
Response to Reviewer 2 Comments
Thank you very much. We appreciate the time and effort taken to provide feedback on our manuscript. We are grateful for the insightful comments and valuable suggestions for improving the article. Following are the responses to the viewer’s comments.
Minor Issues
Point 1: Which is the real added value of using AR in this education scenario? Why could a simple web interface not provide the same information to students? Please motivate better the rationale behind your choice about using AR.
Response 1: A mobile-based AR experience can be realized through web services or custom applications. The web-based application can access AR systems on the internet thus avoiding the hassle of installing the app on mobile and updating it in the future as they are not native to a particular system. But web-based apps require active internet all the time to run. In contrast to web apps, the custom mobile apps need to be installed on the phone but they can work offline without requiring the active internet. Furthermore, mobile apps are faster and more efficient as all computations are performed locally on the device and no external communication over the internet is needed. Speed and the ability to work offline are two crucial factors that forced authors to design an AR system based on mobile.
Point 2: The aim of the work is clear and scientifically relevant, however as the authors stated from students’ feedback, the AR application can eat up a lot of lecture time. Because of this drawback, have my doubts that the application can be effectively used during lectures. Perhaps one solution could be to integrate traditional teaching materials with the content of the app to save time or ensure that students can use it for self-learning outside of lesson times. It would be interesting to know the authors' opinion on this in the discussion Section.
Response 2: We agree with the reviewer that AR4FSM can also be outside the lecture hours. This type of blended learning is already getting popularity as restriction due to pandemic starts easing and universities are already migrating to this pedagogical style where face-to-face lectures are being blended with online activities. This integration of the AR technology with traditional or instructor-led teaching will allow the students to enjoy the best of both worlds and caters to the need of all types of learners. The instructor-led portion of teaching will allow learners to engage easily while the online portion allows managing the pace of learning.
Point 3: Criticism of the interface: seeing the buttons in Figure 9 (a) and (b), there does not seem to be any visual feedback on the state change except a piece of textual information seen in foreshortening. In fact, in lines 386-387 of page 11, authors wrote “the name of the destination state is displayed on the screen to make it more user-friendly and easier to follow.” However, am not very convinced that this solution is easier to follow. Wouldn’t it be better to accentuate this feedback by, for example, inserting a color code? It could be used for the buttons in the same way as for the symbols representing states by giving more value to the AR.
Response 3: We appreciate the suggestion and this will be incorporated into the newer version of the application. But, the current implementation also provides auditory feedback in the form of music. A piece of music is generated when the avatar makes a transition from the current state to the next state. Once it reaches the destination state, distinctive music is generated. In short, feedback is available in the form of text, avatar movement, and music.
Point 4: In lines 418-419 of page 12, the authors wrote “Students were given tasks to complete in order to evaluate the app’s features, such as bit stream pattern identification using FSM.” A more detailed description of the tasks performed by students and the modality of the experiment conduction should be provided for greater clarity.
Response 4: We are sorry about the unclear description. We have provcided more details The instruction material comprised a printed handout which contained a 3-state FSM for recognizing a bit stream having a length of two bits similar to the one shown in Figure 2. Students were given a short demonstration prior to the experiment and went through the whole process of scanning the FSM and changing the input resulting in state transition which could be visualized through text, avatar movement, and music. Different sounds are generated when the avatar is making a transition from one state to another state and when it reaches the destination state. This procedure is similar to one described in subsection 3.7 explaining the application feature. Once a participant has experienced the AR4FSM by accomplishing the task of traversing different states of the FSM, they were asked to fill out a paper-based questionnaire.
Point 5: As for the experiment, authors decided to use a subjective questionnaire consisting of 6 questions. Are such questions standard? There is no indication of this. If so, specify where they were taken from in detail possibly adding references. If they are not standard, think that the user study could negatively affect the reliability of the results obtained. In this case, explain in a convincing way why they were not taken from standard ones generally used in the literature such as SUS, TAM. It is important.
Response 5: These are standard questions taken from TAM3 to measure the rate of students ‘acceptance to use the AR4FSM. We have selected and adopted questions presented by (Ahmad Fauzi, Ali & Amirudin 2019). We have omitted simple technology-related questions considering that these are not applicable to engineering students and especially, to ECSE students who are tech savvy and highly experienced.
Point 6: Regarding the questionnaire, with a view to using the application in an educational context, I think a missing question appears to be "is it easy to learn how to use the application?". Based on the feedback received, did the students understand right away how the application works, or did anyone have difficulty? If so, wouldn’t it be better to include a video tutorial as well? I think such considerations can strengthen the discussion Section.
Response 6: We think this correlates to question 1, “The application was easy to use.” This should be kept in mind that participants had no prior exposure to the application. They were briefed on the spot and all of them were able to complete the task. We can safely assume that it was easy to use. At the same time, we believe that this approach is biased as the researchers and participants share the same background. Therefore, some assumptions made might not be valid for the participant having a different background.
Point 7: In lines 535-537 of page 15, the authors wrote “In the case of AR4FSM, students were able to not only see the FSM but also predict the avatar's future move, making the learning material more enjoyable and gamified.” Where does this statement come from? Did these considerations derive from experimenters’ observations? Specify it, because to have scientific validity such statement must be derived from, for example, a retention test as rightly mentioned in future work.
Response 7: Sorry for the unclear statement. This is a derived statement that comes from the ARCS (Attention, Relevance, Confidence, Satisfied) model presented by Keller. For any FSM drawn on a paper, based on the current state and the status of inputs, the student can make a prediction about the next state using his knowledge but there is no way to see it happening and get feedback. When using AR4FSM, students can change inputs and then visualize the transition of the avatar from the current state to the predicted state. Avatar's movement from one state to another based on input selection gave immediate feedback by confirming their input option. If the avatar proceeds to the projected next stage along the predicted path, it will affirm his understanding and he feels which has a good impact on the student’s motivation. No need to mention that AR is modern technology that immediately grasps the attention of the students and control over learning provides confidence. All these factors contribute to motivating the student.
Point 8: In lines 563-569 of page 15, the authors pointed out the criticality of using a small display which proved to be a barrier to reading. Authors solved the issue by restricting the number of FSM states under investigation, but hwat about using another devices such as table with a large display. Could the problem be solved.
Response 8: We agree with reviewer’s comment that this problem could also be solved using tablets which have bigger displays as compared to mobiles but this is the only benefit they offer. Tablets are a less portable version of s smartphones but they are less commonly used for messaging and calling. Smartphones on the other hand offer all those functionalities offered by the tablet but it is much easier to carry them. Another option could be the use of fixed screen AR technologies such as desktop computers as a two-dimensional medium on monitor displays, but these displays cannot be used in traditional lecture environments and can only be used either in lab setups or small classrooms where computers are available to all students.
Structured Organization
Point 1: In lines 62-63 of page 2, authors presented some AR application fields. It is recommended to anticipate this list so that authors can pass from the general to the particular regarding the educational conext up to the list of the advantages and disadvantages for teacher and students.
Response 1: We agree with reviewers’ comment. We have made the suggested modifications in the revised manuscript.
Point 2: The intorduction seems to be too long. It is suggested to reorganize the contents by separating the literature review section. Doing so, the lines 210-223 should remain in the introduction section. Furthermore, it not needed to talk about the tacking techniques and the display technologies. Suggest removing them from the Introduction and simplifying justify the choice in section 2. Also, the figure about the Milgram continum is no more needed in this type of wok.
Response 2: We agree with the reviewer’s comment. We have reduced the length of the introduction section, created a new section to describe existing work, and removed the figure about the Milgram continuum. We have not only shortened the discussion on racking and display technologies but also moved them to Section 3 where technical aspects are discussed.
Point 3: It is not necessary to explain at the beginning of each section what will be presented as for lines 225, 399-401, 465-467m 506-509, rather at the end of the introduction section there should be a summary of the sections presented in the work in addition to that of the case study with the tests performed.
Response 3: We agree with the reviewer’s comment and these changes have been incorporated in the revised manuscript.
Point 4:
Response 3: We agree with the reviewer’s comment and considered the recommended research to address the issue.
Writing and Reading
Thank you very much for pointing out these issues. We have fixed all these issues (point 1 – point 15) related to reading and writing in the revised manuscript.
Point 1: In lines 52-53 of page 2, authors provided the definition of AR without explaining what is meant by Mixed Reality shown in Figure 1. Please fix it.
Point 2: In line 177 of page 4, the plural should be used for the word “author”.
Point 3: The resolution of figure 4 should be improved. It is also recommended to indicate each figure with the letters (a), (b) and (c).
Point 4: In lines 349-350 of page 9, authors wrote “Instructions are also provided on how to concatenate and invert conditions, as shown in Figure 8.” However, the button that provides access to these instructions is not clear. Is it perhaps the “help” command seen in Figure 6 (b)? Please clarify it.
Point 5: It is suggested to always use the quotation marks in cases such as “and” keyword, “not” keyword, and “not” condition.
Point 6: In lines 371-372 of page 10, authors wrote “the user can toggle the values between “0” and “1” by simply tapping on the corresponding input name.” However, no figure shows this. It is suggested to modify the Figure 9 (a) adding this detail.
Point 7: Figure 10 seems to have something wrong because in (a) the avatar seems to be going from state 1 to 3, while in (b) there are conditions. Please check them.
Point 8: In line 411 of page 12, authors wrote “A total of 60 students from the ECSE department (age 19-22) took part in the study. It should be added the information about SD, average age, and percentage of female sex even if shown after in Table 2.
Point 9: It is advisable to always use abbreviations after the mentionings, e.g., AR.
Point 10: In line 417 of page 12, authors wrote “AR4FS” instead of “AR4FSM”. Please fix it.
Point 11: Lines 432-444 of page 12 should be cut out to avoid repetitions and make the reading smoother.
Point 12: In Table 2, it is not clear why in the total there is 57 with 100% both for designations and degree instead of 60, is this an error? Please fix it.
Point 13: In line 517 of page 14, authors wrote “finding s” instead of “findings”. Please fix it.
Point 14: In line 584-585 of page 16, authors wrote about the possibility to explore the students’ learning for future work. However, it seems to be a repetition of the intention to do a retention test expressed in lines 579-580. Please fix it.
Point 15: It is recommended to check the correct formatting of paragraphs.
Response (1-15)
Thank you very much for pointing out these issues. We have fixed all these issues (point 1 – point 15) in the revised manuscript.
Reviewer 3 Report
I have reviewed the manuscript ID (education-1792341) entitled “AR4FSM: Mobile Augmented Reality Application in Engineering Education for Finite State Machine Understanding”, the manuscript has a good idea and contribution. The quality of the research work presented in the study is good, and the paper is scientifically sound, with well-modified results. I recommend that it be accepted for publication.
Author Response
Thank you very much. We appreciate the time and effort taken to provide feedback on our manuscript. We are grateful for the insightful comments and valuable suggestions for improving the article. We have done an extensive revision and incorporated most of the suggestions made by the reviewers. The changes made to the text have been highlighted in the revised manuscript.
Reviewer 4 Report
The findings in lines 488-504 have not any percentages
and they did not consider gender issues
Author Response

(The authors gave the same response as above.)

Round 2
Reviewer 1 Report
The manuscript has improved notably. I thank the author(s) for the effort. One final comment: I think the manuscript needs a revision in terms of English grammar.
Reviewer 2 Report
I would thank the authors for having improved their manuscript following my suggestions.